# 70% Size, 100% Accuracy:
# Lossless LLM Compression for Efficient GPU Inference via Dynamic-Length Float (DFloat11)

**Tianyi Zhang**[1], **Mohsen Hariri**[2], **Shaochen (Henry) Zhong**[1], **Vipin Chaudhary**[2], **Yang Sui**[1], **Xia Hu**[1], and **Anshumali Shrivastava**[1,3]

[1]Department of Computer Science, Rice University
[2]Department of Computer and Data Sciences, Case Western Reserve University
[3]Ken Kennedy Institute

{tz21, henry.zhong, yang.sui, xia.hu, anshumali}@rice.edu, {mohsen.hariri, vipin}@case.edu

Code: https://github.com/LeanModels/DFloat11
Models: https://huggingface.co/DFloat11

## Abstract

Large-scale AI models, such as Large Language Models (LLMs) and Diffusion Models (DMs), have grown rapidly in size, creating significant challenges for efficient deployment on resource-constrained hardware. In this paper, we introduce *Dynamic-Length Float* (DFloat11), a lossless compression framework that reduces LLM and DM size by 30% while preserving outputs that are bit-for-bit identical to the original model. DFloat11 is motivated by the low entropy in the BFloat16 weight representation of LLMs, which reveals significant inefficiency in the existing storage format. By applying entropy coding, DFloat11 assigns dynamic-length encodings to weights based on frequency, achieving near information-optimal compression without any loss of precision. To facilitate efficient inference with dynamic-length encodings, we develop a custom GPU kernel for fast online decompression. Our design incorporates the following: (i) compact, hierarchical lookup tables (LUTs) that fit within GPU SRAM for efficient decoding, (ii) a two-phase GPU kernel for coordinating thread read/write positions using lightweight auxiliary variables, and (iii) transformer-block-level decompression to minimize latency. Experiments on Llama 3.3, Qwen 3, Mistral 3, FLUX.1, and others validate our hypothesis that DFloat11 achieves around 30% model size reduction while preserving bit-for-bit identical outputs. Compared to a potential alternative of offloading parts of an uncompressed model to the CPU to meet memory constraints, DFloat11 achieves 2.3–46.2× higher throughput in token generation. With a fixed GPU memory budget, DFloat11 enables 5.7–14.9× longer generation lengths than uncompressed models. Notably, our method enables lossless inference of *Llama 3.1 405B*, an 810GB model, on a single node equipped with 8×80GB GPUs.

## 1 Introduction

Foundation models, such as Large Language Models (LLMs) and Diffusion Models (DMs), have demonstrated remarkable capabilities across a wide range of Natural Language Processing (NLP) [56] and Computer Vision (CV) tasks [57]. However, their huge model sizes create substantial obstacles

39th Conference on Neural Information Processing Systems (NeurIPS 2025).

for efficient deployment, especially in memory-constrained environments. For example, a competitive recent LLM, *Llama 3.1 405B* [20], has 405 billion parameters in 16-bit Brain Float (BFloat16) format and requires about 810 GB of memory for full inference, exceeding the capacity of a typical high-end GPU server (e.g., DGX A100/H100 with $8 \times 80$GB GPUs). As a result, deploying this model requires multiple nodes, making it expensive and inaccessible. **In this work, we present a solution that compresses any BFloat16 model to approximately 70% of its original size while preserving 100% of its accuracy on any task.**

**Model compression via quantization has limitations.**   Quantization is a type of *lossy* compression method that lowers the precision of model weights by converting them into lower bit-width representations [15, 37, 36, 43]. Although it can significantly reduce memory usage and often improve inference speed, quantization is not a one-size-fits-all solution and presents several key limitations: ❶ *Accuracy degradation*. By design, quantization introduces approximation errors. The degree of accuracy loss depends on multiple factors, including the base model, quantization method, evaluation benchmark, and target bit-width [35]. These interactions make it difficult to predict or quantify the impact comprehensively. Even mild quantization can noticeably degrade performance. For example, applying 8-bit SmoothQuant [51] to *DeepSeek-R1-Distill-Qwen-1.5B* [21] results in a 9.09% drop in average accuracy across reasoning tasks [39]. ❷ *Behavioral shifts.* Even when overall accuracy metrics appear roughly unchanged, quantized models may behave differently from their full-precision counterparts. For instance, Dutta et al. [13] observe a phenomenon called *flips*, where quantized models produce answers that change from correct to incorrect and vice versa. This indicates that quantization can significantly alter model behavior, even when standard accuracy metrics show minimal change. For example, the W8A16 GPTQ-quantized Qwen2-1.5B[15, 54] exhibits only a 0.3% drop in GSM8K (8-shot) accuracy [5], yet 6.37% of its answers flip in correctness [13]. ❸ *Compliance and reliability concerns.* In domains like finance or healthcare, quantized models may not satisfy regulatory or reliability standards, as their outputs may differ from those of the original models [31]. We refer readers to Appendix A for a more detailed discussion on quantization.

**Existing lossless model compression does not support efficient GPU inference.**   Unlike lossy compression, *lossless compression* reduces model size while preserving the full precision of the original weights. This ensures the model's output distribution remains identical to that of the uncompressed counterpart. However, most existing lossless methods focus on storage efficiency, such as compressing model checkpoints [22, 25], or target specialized hardware like FPGAs [59], rather than accelerating inference on general-purpose GPUs. While useful for tasks like checkpoint rollback during large-scale training [47] or reducing download time from model hubs [25], these methods offer little to no benefit for GPU-based inference.

**Our proposal, Dynamic-Length Float (DFloat11), is a lossless compression framework optimized for efficient GPU inference.**   We identify a key inefficiency in the commonly used BFloat16 format: its 8-bit exponent field carries only about 2.6 bits of actual information. This redundancy is consistent across a wide range of LLMs, as shown in Section 2.2. To exploit it, we apply Huffman coding [28] to the exponent bits of BFloat16 weights, while leaving the sign and mantissa bits uncompressed. The resulting exponents have dynamic-length encodings: frequent values are assigned shorter codes, while rarer ones use longer codes. However, standard Huffman decoding relies on sequential bit-by-bit tree traversal, which is inefficient on GPUs due to limited parallelism. Assigning one GPU thread per decompression task leads to severe hardware underutilization and high latency. To overcome this, we design a hardware-aware algorithm that enables efficient online decompression of dynamic-length floats on GPUs. Our solution includes three key components: 1. compact, hierarchical lookup tables (LUTs) that fit in GPU SRAM to support fast, table-based Huffman decoding, 2. a two-phase GPU kernel that uses lightweight auxiliary variables to coordinate thread-level read and write operations, and 3. batched decompression at the transformer-block level to maximize throughput. We summarize our contributions as follows:

1. We propose **Dynamic-Length Float (DFloat11)**, a losslessly compressed floating-point format that reduces BFloat16 weights to approximately 11 bits. This yields around 30% model size reduction with bit-for-bit identical outputs.

2. We develop optimized, hardware-aware algorithms for efficient GPU inference with DFloat11-compressed models by leveraging GPU memory and compute hierarchies.

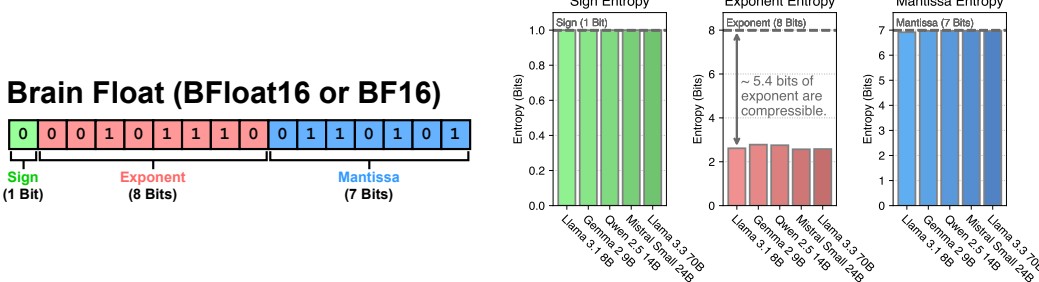

Figure 1: **(Left)** The allocation of bits for the components of BFloat16. **(Right 3)** The Shannon entropy of the components (sign, exponent, mantissa) of BFloat16 weights in various LLMs.

3. We evaluate DFloat11 across popular LLMs and diffusion transformers, including Llama 3, Qwen 3, Mistral 3, DeepSeek R1 Distilled, FLUX.1, and Stable Diffusion 3.5 [20, 46, 45, 21, 32, 2]. Our method consistently achieves 30% compression without altering original outputs at all. Notably, it enables running *Llama-3.1-405B on a single node* ($8 \times 80$GB A100 GPUs), reducing hardware requirements by half without accuracy loss.

## 2 Method

In this section, we introduce our proposed floating-point format, Dynamic-Length Float (DFloat11), along with its custom decompression kernel designed for efficient GPU inference.

### 2.1 Preliminary

**Brain Float (BFloat16)** Recent state-of-the-art LLMs predominantly employ the 16-bit Brain Float format (BFloat16 or BF16) for storing weights, due to its balance of numerical precision with memory efficiency. BF16 allocates its 16 bits as follows: 1 *sign* bit, 8 *exponent* bits, and 7 *mantissa* bits. The numerical value represented by a BF16 number is computed as:

$$(-1)^{\text{sign}} \times 2^{\text{exponent}-127} \times (1.\text{mantissa}), \tag{1}$$

where $\text{mantissa}$ is interpreted as a binary fractional value.

**Entropy Coding** Entropy coding is a core technique in lossless data compression that leverages statistical redundancy to reduce data size. Several widely used methods fall under this category, including Huffman coding [28], arithmetic coding [33], and Asymmetric Numeral Systems (ANS) [12]. Among these, Huffman coding is one of the most widely adopted, which uses variable-length encoding to minimize the size of encoded data. It assigns shorter binary codes to more frequent symbols and longer codes to less frequent ones. The codes are decoded using a prefix-free binary tree, known as a Huffman tree. Due to the prefix-free property of Huffman codes, no code is a prefix of any other, which ensures unique decodability of the encoded bitstream without the need for delimiters. The tree is constructed based on symbol frequencies and is provably optimal for any given frequency distribution. However, decoding Huffman codes in a massively parallel manner is challenging due to its inherently sequential nature.

**GPU Computation and Memory Paradigm** GPUs are designed to perform computations in a massively parallel manner. A modern GPU consists of thousands of threads, which are organized into blocks and executed on streaming multiprocessors (SMs). Each block has access to a small, fast, on-chip memory called shared memory (often referred to as SRAM), which provides much lower latency and higher bandwidth than the off-chip global memory, commonly known as high-bandwidth memory (HBM). The capacity of shared memory is limited, typically having up to 100 KB per block. In this work, we leverage the fast access characteristics of SRAM to enable efficient on-the-fly decompression of compressed weights during inference.

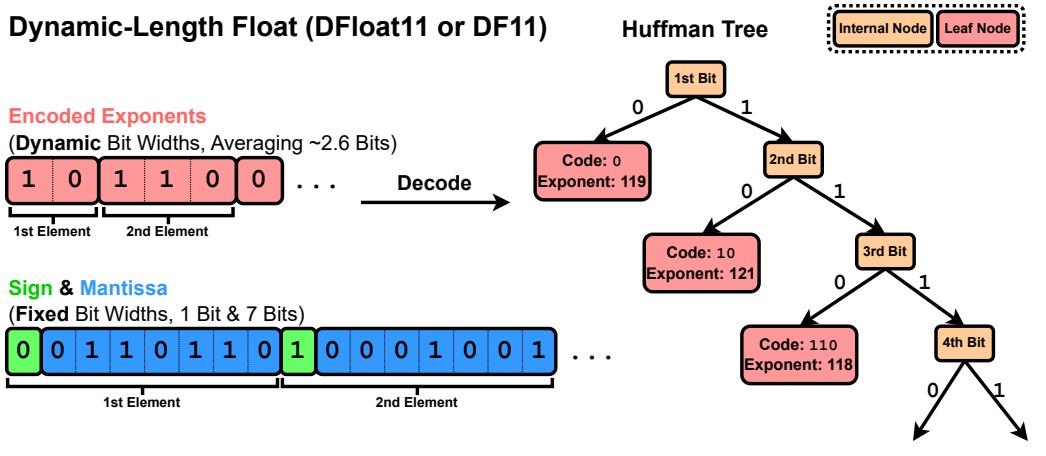

Figure 2: Our proposed format *Dynamic-Length Float* for compressing BFloat16 weights of LLMs losslessly down to 11 bits. The exponents are compressed via Huffman coding, while the sign and mantissa bits remain uncompressed.

## 2.2 Motivation: BFloat16 Representation is Information Inefficient

To motivate the lossless compression of LLM weights, we analyze the compressibility of the BFloat16 weights of recent LLMs. Specifically, we use Shannon entropy to quantify the information content of BFloat16 components (sign, exponent, and mantissa) for all linear projection matrices of an LLM. The Shannon entropy $H(\cdot)$ is defined as:

$$H(X) = -\sum_{x \in \mathcal{X}} p(x) \log_2 p(x) \tag{2}$$

where $X$ is a discrete random variable with support $\mathcal{X}$, and $p : \mathcal{X} \to [0, 1]$ denotes its probability mass function. We present the computed entropy values in Figure 1. As shown, the entropy of the sign and mantissa bits is close to their respective bit widths, indicating limited potential for compression. In contrast, the exponent exhibits significantly lower entropy, approximately 2.6 bits versus its allocated 8 bits, suggesting substantial opportunities for lossless compression.

To understand this discrepancy, we visualize the frequency distribution of all BFloat16 components in Figure 8 and the ranked frequency of exponent values in Figure 9, both in the Appendix. The sign and mantissa values are relatively uniform across their ranges, but the exponent distribution is highly imbalanced: only about 40 of the 256 possible 8-bit values are used, with the rest never appearing. Ranked frequencies also decay rapidly. These observations reveal the low entropy of the exponent and its potential for compression.

## 2.3 Dynamic-Length Float: Lossless LLM Compression for Efficient GPU Inference

To address the substantial information inefficiency in the BFloat16 representation of LLM weights, we propose a lossless compression framework that encodes floating-point parameters using entropy coding. Specifically, we build a Huffman tree based on the distribution of exponents in model weights. We then compress the exponents using Huffman coding, while preserving the original signs and mantissas. Exponents are encoded and tightly bit-packed into a byte array, EncodedExponent, while the sign and mantissa are left uncompressed and stored in a separate byte array PackedSignMantissa. Figure 2 illustrates Dynamic-Length Float (DFloat11 or DF11), our proposed format for compactly representing BFloat16 model parameters.

**The Core Challenge: Efficient GPU Inference with Compressed Weights**   While DFloat11 enables lossless compression of LLM weights, efficient GPU inference remains a key challenge. Entropy-coded weights use variable-length encoding and cannot be directly used in matrix multiplications. As a result, each weight matrix must be decompressed on-the-fly to its original BFloat16 format

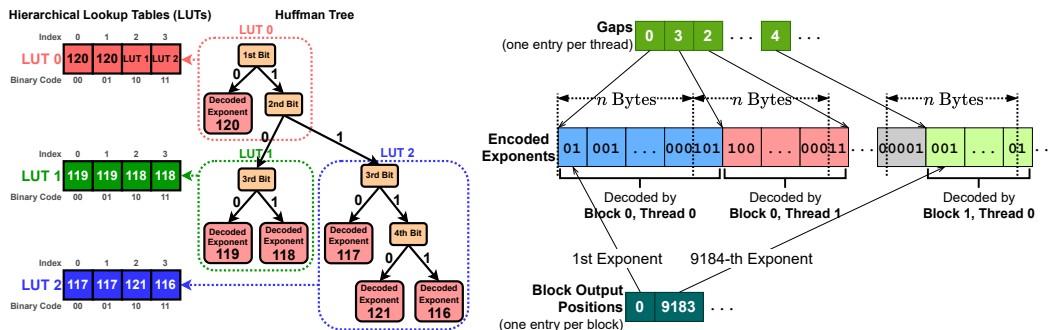

Figure 3: **(Left)** The Huffman tree is decomposed into a set of non-overlapping subtrees, each corresponding to a compact lookup table (LUT). These hierarchical LUTs reside in GPU SRAM to enable efficient Huffman decoding via array lookups. **(Right)** Each thread decodes $n$ bytes of encoded exponents. The array *Gaps* stores the bit offset of the first element assigned to each thread, while the array *Block Output Positions* stores the index of the first element for each thread block.

before matrix multiplication, then discarded immediately after use to conserve memory. However, traditional Huffman decoding is inherently sequential, requiring bit-by-bit tree traversal for each element, which is ill-suited for GPUs' parallel architecture. Naively assigning a single thread for decompression leads to poor utilization and high latency. Addressing this bottleneck is essential for practical compressed inference.

In the following paragraphs, we present our solution in detail: a set of hardware-aware algorithmic designs tailored for low-latency decoding of entropy-coded weights in a massively parallel manner. Our approach consists of three key components: ❶ leveraging compact lookup tables that fit within GPU SRAM for efficient, lookup-based decoding, ❷ introducing a two-phase kernel design to coordinate read/write operations for all threads using lightweight auxiliary variables, and ❸ performing decompression at the transformer block level to minimize latency.

### 2.3.1 Efficient Decoding with Hierarchical Lookup Tables

The traditional approach to decoding Huffman codes involves reading the encoded bitstream bit by bit and traversing the Huffman tree accordingly. However, this method is inefficient on GPUs due to frequent branching and limited parallelism. To enable efficient decoding on GPUs, we adopt a lookup-table-based approach [53].

Assume the maximum Huffman code length is $L$, and we construct a lookup table LUT of size $2^L$. At each index $i$, LUT stores the decoded exponent whose Huffman code matches the prefix of the binary representation of $i$. To decode the next exponent, we read the next $L$ bits from the encoded bitstream, interpret them as an index into LUT, and retrieve the corresponding value. To determine how many bits to advance in the stream, we use a secondary lookup table CodeLengths, which maps each exponent to the length of its Huffman code. A detailed example of this decoding process is provided in Section I of the Appendix.

In practice, the value of $L$ can be large. For LLMs, $L$ typically ranges from 24 to 32, resulting in a LUT with up to $2^{32}$ entries, which cannot fit within GPU SRAM for fast lookups. To address this, we decompose the monolithic LUT into a hierarchy of compact lookup tables [53]. We first partition the Huffman tree into non-overlapping subtrees of height 8. Each subtree corresponds to a compact LUT that decodes 8 bits, requiring only $2^8 = 256$ entries.

Figure 3 shows an example of how a Huffman tree of height 4 can be decomposed into a hierarchy of compact LUTs, each with 4 entries. Because the LUTs are organized hierarchically, some entries must serve as references to other LUTs lower in the hierarchy. We take advantage of the sparsity in 8-bit exponent usage: although 256 values are available, typically only around 40 are used in LLMs (see Figure 9 in the Appendix). We repurpose unused values (specifically, the range 240 to 255) as pointers to other LUTs. These values correspond to extremely large magnitudes ($\pm 2^{113}$ to $\pm 2^{128}$) that do not occur in LLM weights, making them safe for use as internal markers.

We use $k$ to denote the number of compact LUTs. In our experiments, we observe that $k$ ranges from 4 to 8 for the Huffman trees built from BFloat16 exponent values. Combined with CodeLengths, these LUTs occupy at most $(8 + 1) \times 256$ bytes of memory, which easily fits within SRAM and allows for fast repeated lookups.

### 2.3.2  Two-Phase Kernel and Lightweight Auxiliary Variables

To leverage the parallel processing capabilities of GPUs, we assign each thread to a contiguous, non-overlapping block of encoded exponents consisting of $n$ bytes ($n = 8$ in our experiments). Each thread decodes elements whose Huffman codes begin within its assigned block. Since Huffman codes are variable-length, a thread may need to skip some bits at the start before decoding the first element. Similarly, the last element may span beyond the assigned byte range.

This approach introduces two key challenges: 1. The starting bit position for each thread is unclear due to the variable-length nature of Huffman codes. 2. Except for the first thread, the index of decoded elements is unknown, making it difficult to determine their correct output locations.

To address the first issue, we use a gap array [53] to specify the starting bit offset for each thread. The array Gaps has one entry per thread, where each entry indicates the offset of the first valid Huffman code relative to the thread's assigned starting byte. With a maximum code length of 32 bits, each offset lies in $[0, 31]$ and is stored using only 5 bits.

For the second issue, maintaining an output position for each thread is straightforward but memory-intensive. Each position requires a 32-bit integer, and with tens of thousands of threads per weight matrix, this leads to significant overhead, undermining DFloat11's compression benefits. To reduce this overhead, we store the output position only for the first element of each thread block rather than for every thread. Since each block typically contains hundreds to thousands of threads, this optimization reduces the overhead from one 32-bit integer per thread to one per block, making the memory cost negligible. Figure 3 illustrates how the *gap* and *block-level output position* arrays encode the metadata associated with the encoded exponents.

To support this design, we implement a *two-phase* kernel. In the **first phase**, each thread decodes its assigned block and counts the number of elements, without writing to the HBM. Afterward, threads within a block synchronize to compute per-thread output positions via a prefix sum over the element counts. We use the Blelloch algorithm [4] for this step. In the **second phase**, each thread re-decodes the same block, this time writing decoded values to a write buffer in SRAM at the calculated positions. To avoid redundant global memory access, the encoded exponents are loaded into SRAM before the first pass. Once all decoded exponents are written to SRAM, a single batch of coalesced writes is issued to HBM. Pseudocode for the two-phase kernel is provided in Algorithm 1 of the Appendix.

### 2.3.3  Transformer-Block-Level Decompression

We now have a complete recipe for decompressing entropy-coded exponents in a massively parallel manner. During inference, the LLM weights stored in DFloat11 format, along with auxiliary variables (the thread-level gap array and block-level output position array), reside entirely in GPU memory. When a weight matrix is needed for matrix multiplication, it is decompressed on-the-fly into the original BFloat16 format. Once the matrix multiplication is complete, the BFloat16 matrix is immediately discarded to conserve GPU memory.

In practice, decompressing a single weight matrix often underutilizes GPU resources due to its relatively small size. As the matrix size increases, decompression throughput improves. Figure 7 illustrates this trend, showing how DFloat11 decompression scales with matrix size. To capitalize on this, we propose batching the decompression of multiple matrices together to improve throughput and hide latency. Specifically, we decompress all DFloat11 weight matrices within a transformer block as a single batch. This batched decompression occurs right before the forward pass of the transformer block. We also compress the token embedding and language modeling head of LLMs. Since these matrices are large enough to saturate GPU resources, batching their decompression is unnecessary.

Table 1: DF11 statistics for various models. Model sizes are shown before and after compression.

| Model | Original → DF11 Compressed | Compression Ratio | Avg. Bit Width |
|---|---|---|---|
| *Large Language Models* | | | |
| Llama 3.1 8B Instruct | 16.06 GB → 10.90 GB | 67.84% | 10.85 |
| Llama 3.3 70B Instruct | 141.11 GB → 95.40 GB | 67.61% | 10.82 |
| Llama 3.1 405B Instruct | 811.71 GB → 551.22 GB | 67.91% | 10.87 |
| Qwen 3 14B | 29.54 GB → 20.14 GB | 68.17% | 10.91 |
| QwQ 32B | 65.53 GB → 44.65 GB | 68.14% | 10.90 |
| Mistral Nemo Instruct | 24.50 GB → 16.59 GB | 67.74% | 10.84 |
| Mistral Small 3 | 47.14 GB → 31.86 GB | 67.58% | 10.81 |
| Phi 4 Reasoning Plus | 29.32 GB → 19.83 GB | 67.64% | 10.82 |
| DeepSeek R1 Distill Llama 8B | 16.06 GB → 10.89 GB | 67.81% | 10.85 |
| *Diffusion Transformers* | | | |
| FLUX.1 dev | 23.80 GB → 16.33 GB | 68.61% | 10.98 |
| FLUX.1 schnell | 23.78 GB → 16.31 GB | 68.58% | 10.97 |
| Stable Diffusion 3.5 Large | 16.29 GB → 11.33 GB | 69.52% | 11.12 |

Table 2: Comparison of accuracy and perplexity for the BF16 and DF11 models on different benchmarks. DF11 compression results in absolutely no loss in accuracy or perplexity.

| Model | Data Type | Accuracy | | Perplexity | |
|---|---|---|---|---|---|
| | | MMLU | TruthfulQA | WikiText | C4 |
| Llama 3.1 8B Instruct | BF16 | $68.010 \pm 0.375$ | $36.965 \pm 1.690$ | 8.649 | 21.677 |
| | DF11 (Ours) | $68.010 \pm 0.375$ | $36.965 \pm 1.690$ | 8.649 | 21.677 |

## 3 Experiments

We empirically evaluate the effectiveness of DF11 compression and its GPU inference efficiency. A range of recent LLMs and DMs are compressed from their original BFloat16 format into DF11, and we report the resulting compression ratios. We then compare the inference performance of DF11-compressed models against their uncompressed counterparts across different GPUs, followed by an ablation study to analyze the impact of compression.

**Software and Hardware** We implement the DF11 decompression kernel in CUDA and C++, and integrate it into the HuggingFace Transformers [48] inference framework. We evaluate the inference efficiency of our DF11 models against the original BF16 counterparts. We use the HuggingFace Accelerate framework to support CPU offloading and multi-GPU inference. To assess the performance of the DF11 kernel across different hardware configurations, we run experiments on multiple machines with varying GPU and CPU setups. The hardware specifications for all experimental machines are provided in Table 4 in the Appendix.

### 3.1 Results

**DF11 compresses models to 70% size.** Table 1 presents the compression factors of DF11 for a wide selection of recent LLMs and DMs. Specifically, we apply compression to all weight matrices and token embeddings in LLMs and all weight matrices in the transformer blocks of DMs. The models we compress include Llama 3.1/3.3 [20], Qwen 3 [54], Mistral Nemo/Small [44, 45], Phi 4 [1], DeepSeek R1 Distilled [21], Stable Diffusion 3.5 [2], FLUX.1 [32]. DF11 achieves approximately 70% compression across all models, corresponding to an effective bit width of around 11 bits.

**Accuracy and perplexity evaluations confirm DF11 compression is lossless.** We verify the lossless property of DF11 compression through a series of accuracy and perplexity evaluations on standard benchmarks. Evaluations are conducted using `lm_evaluation_harness` [18], reporting accuracy on MMLU [24] and TruthfulQA [38], and word-level perplexity on WikiText [41] and C4 [42]. The results are shown in Table 2. As demonstrated, the compressed model achieves identical accuracy and perplexity to the original BF16 counterpart. We also present the text-to-image

Table 3: Comparison of peak GPU memory usage and text-to-image generation time for diffusion transformers in BF16 and DF11, using a single A5000 GPU.

| | Peak GPU Memory (GB) | | Generation Time (s) | |
|---|---|---|---|---|
| Model | BF16 | DF11 (Ours) | BF16 | DF11 (Ours) |
| Stable Diffusion 3.5 Large | 16.44 | 11.78 | $66.36 \pm 0.13$ | $69.08 \pm 0.11$ |
| FLUX.1 dev | 23.15 | 16.72 | $74.41 \pm 0.15$ | $78.53 \pm 0.18$ |

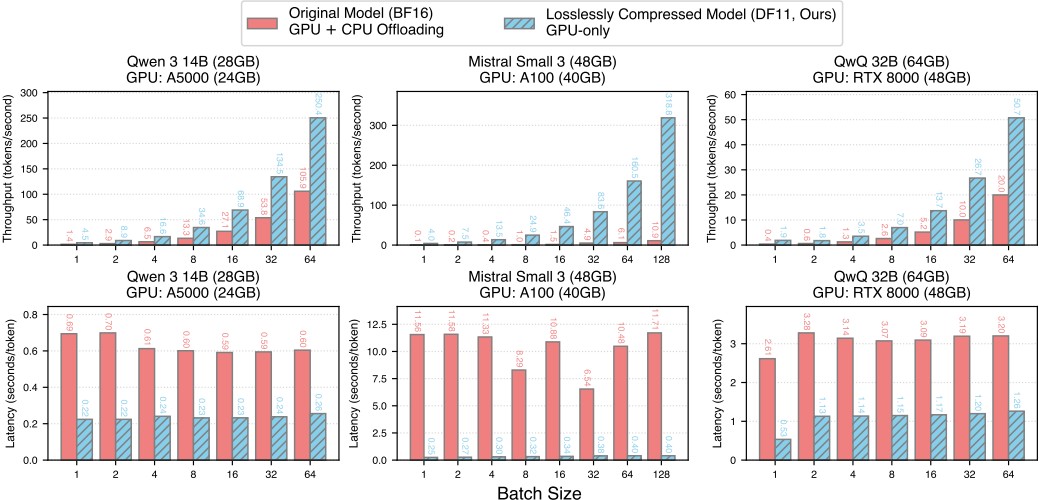

Figure 4: Comparison of throughput (**top row**) and latency (**bottom row**) for token decoding using the original BF16 models and their DF11-compressed counterparts. Portions of the BF16 models are offloaded to the CPU due to GPU memory constraints.

generation results of BF16 and DF11 Stable Diffusion 3.5 Large model in Appendix J. Given the same random seed and text prompt, the image generated are pixel-wise identical with the original model.

**DF11 outperforms CPU offloading in inference efficiency.** We compare the inference performance of DF11 and BF16 models across various hardware platforms. Due to memory constraints, BF16 models exceed the capacity of a single GPU and require partial CPU offloading, while DF11 models fit entirely within GPU memory. For fair comparison, we retain most computation on the GPU for BF16 models and offload only necessary components. Latency and throughput are measured after a 100-token warm-up run, followed by decoding 100 tokens from an empty prompt across varying batch sizes. Each configuration is run five times, and we report the average results. As shown in Figure 4, DF11 consistently outperforms BF16 with CPU offloading, achieving 2.31–46.24× lower latency or higher throughput. Multi-GPU comparisons are shown in Figure 10 in the Appendix.

**DF11 reduces memory usage for diffusion transformers with minimal latency impact.** We assess the impact of DF11 compression on diffusion transformer models by measuring peak GPU memory usage and text-to-image generation latency for an 1024×1024 image across five runs. Neither the BF16 nor DF11 models employ CPU offloading. As shown in Table 3, DF11 reduces memory consumption by 28.3% for Stable Diffusion 3.5 and 27.8% for FLUX.1. The relative increase in latency is small: 4.1% for Stable Diffusion and 5.5% for FLUX.1.

**DF11 memory savings enable longer generation lengths.** DF11 compression not only can reduce the number of GPUs needed for inference but can also support longer generation under the same VRAM budget. During decoding, the KV cache grows linearly with the number of tokens and quickly becomes a memory bottleneck. Figure 5 shows GPU memory usage for DF11 and BF16 models with batch size 1 as token count increases. DF11 allows 5.70 to 14.86× more tokens to be decoded before reaching memory limits.

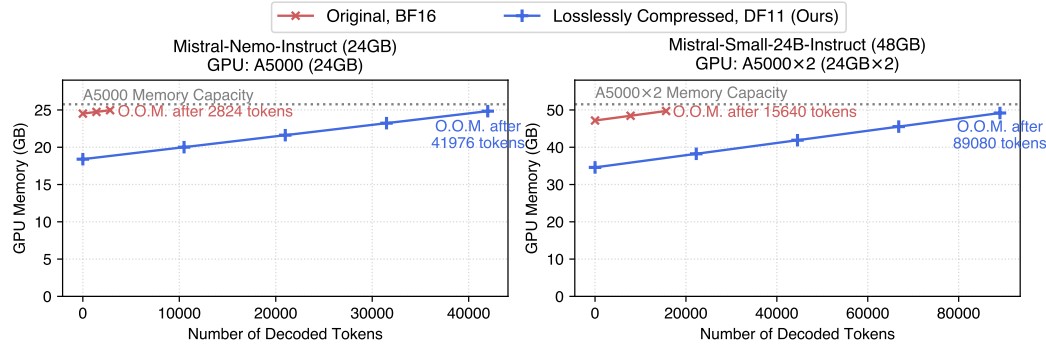

Figure 5: Comparison of GPU memory consumption between BF16 models and DF11 counterparts. The DF11 models support 5.70–14.86× longer context lengths by allowing more GPU memory to be used for storing the KV cache. "O.O.M." means out of memory.

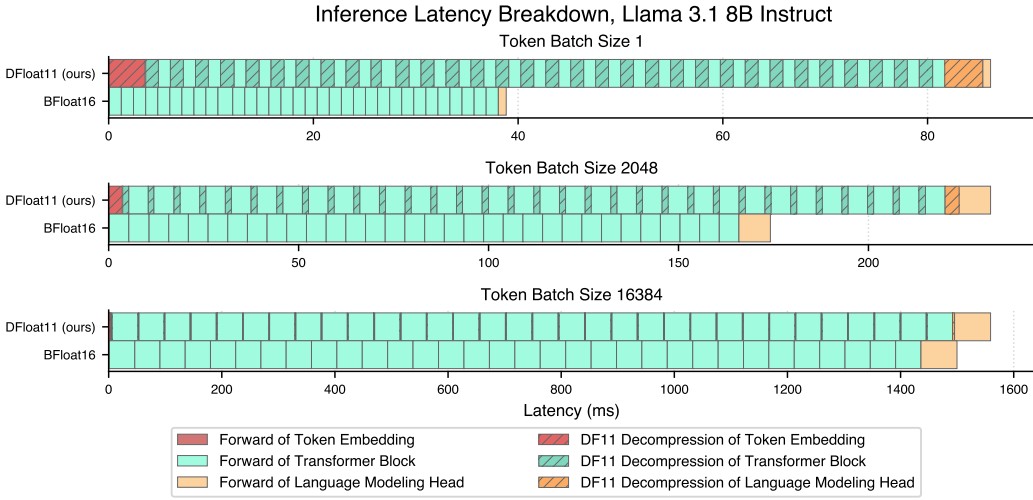

Figure 6: Comparison of latency breakdown for DF11 and BF16 Llama 3.1 8B Instruct during GPU inference for different token batch sizes, using one A100-40GB GPU.

## 3.2 Ablation Study

**Latency breakdown shows decompression overhead is amortized at larger batch sizes.** We analyze the latency of *Llama 3.1 8B Instruct* in BF16 and DF11 formats across varying token batch sizes on an A100-40GB GPU. For each setting, we measure the average latency of each component over 10 runs, as shown in Figure 6. DF11 introduces additional latency from decompressing the token embedding, transformer blocks, and language modeling head. This overhead is constant and independent of batch size, so increasing the token batch size effectively amortizes the cost.

**DF11 decompression is significantly faster than CPU-to-GPU transfer and nvCOMP ANS.** We compare DF11 decompression latency and throughput with two baselines: CPU-to-GPU weight transfer and ANS decompression [12] from NVIDIA's nvCOMP [6], using sliced weight matrices from the Llama 3.1 8B Instruct language modeling head. As shown in Figure 7, DF11 achieves up to 34.95× higher throughput than CPU transfer and up to 20.97× faster decompression than nvCOMP. DF11 also offers a better compression ratio (68%) compared to nvCOMP (79%). Moreover, DF11 decompression throughput improves with larger matrix sizes due to better GPU utilization.

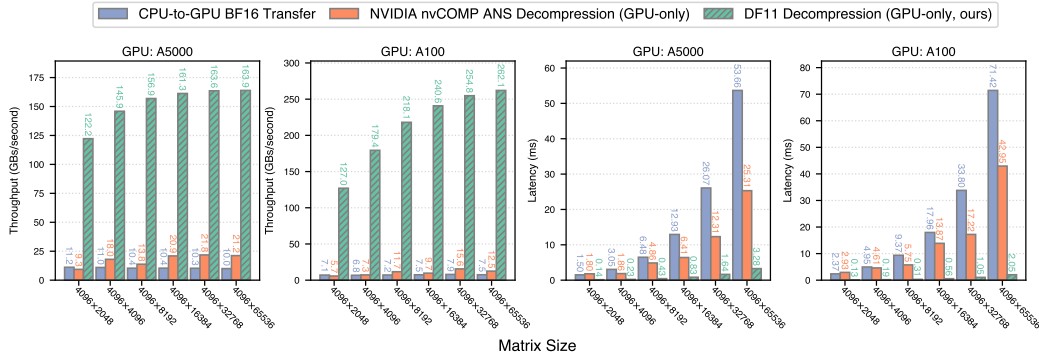

Figure 7: Throughput **(left two)** and latency **(right two)** comparisons between transferring BF16 matrices from CPU to GPU and decompressing the same matrices on GPU using the NVIDIA nvCOMP ANS library and our proposed DF11 kernel, across matrix sizes and GPU types.

# 4   Related Works

**Data Formats for Model Weights** Full-precision model weights are typically stored in formats such as BF16, FP16, or FP32. Several works have proposed 4-bit compressed formats, including FP4, INT4, NF4 (NormalFloat) [9], AF4 (AbnormalFloat) [58], and SF4 (Student Float) [11], which represent each parameter with 4 bits. Unlike these lossy formats, the proposed DF11 format compresses weights losslessly.

**Lossless Model Compression** While lossy compression methods such as pruning [14] and quantization [37, 15] are well-studied, lossless compression remains less explored. Four prior works have addressed this area. *Deep Compression* [22] applied Huffman coding [28] to quantized CNNs, achieving 22% additional compression. *ZipNN* [25] extended this approach to language models with improved compression over classical methods. However, both techniques target storage efficiency and do not support inference-time gains. *NeuZip* [23] is the only prior work supporting GPU inference. It uses Asymmetric Numeral Systems (ANS) with layer-wise decompression and relies on NVIDIA's nvCOMP for GPU-based operations. nvCOMP is no longer open source, and its binary-only distribution limits adoption. Moreover, as shown in Figure 7, nvCOMP ANS incurs higher latency and lower throughput compared to our DFloat11 kernel. *Huff-LLM* [59] is designed for FPGA-like hardware and is not applicable to GPUs. Additional discussion of related formats is presented in Appendix B.

# 5   Conclusion

We introduce *Dynamic-Length Float* (DFloat11), a lossless compression framework designed for efficient GPU inference of BFloat16 models, including both large language models (LLMs) and diffusion models (DMs). DFloat11 exploits the information redundancy inherent in foundation model weights through entropy-coded, dynamic-length encoding, achieving compression rates close to the information-theoretic limit. To enable efficient deployment, we develop hardware-aware algorithms that support high-speed inference directly on compressed weights. Extensive experiments demonstrate that DFloat11 significantly reduces GPU memory requirements for LLMs and DMs, allowing for longer generation lengths, while maintaining bit-exact accuracy and incurring only negligible decompression overhead.

## Acknowledgements

This work was supported by National Science Foundation SHF-2211815 and Ken Kennedy Institute Cluster Grants. Additionally, Henry and Xia are supported by ITE-2429680, IIS-2310260, and US Department of Transportation (USDOT) Tier-1 University Transportation Center (UTC) Transportation Cybersecurity Center for Advanced Research and Education (CYBER-CARE) grant #69A3552348332. Mohsen and Vipin are supported by OAC-2320952, OAC-2112606, and OAC-2117439. The views and conclusions in this paper are those of the authors and do not represent the views of any funding or supporting agencies.

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

# Appendix

## A    Discussion: Is Quantization a Universal Solution?

Much of the motivation behind our work lies in understanding **whether lossless compression of large-scale models such as LLMs, which preserves 100% identical output behavior compared to the original uncompressed model, is a practical direction worthy of further study.** Specifically, how does DFloat11, which compresses LLMs to approximately 11 bits, compare to widely used lossy quantization techniques [15, 37], where models are typically reduced to even lower bit-widths (e.g., 8-bit or 4-bit)?

The answer is far more nuanced than a simple "Yes/No" or a one-size-fits-all judgment about which approach is better. For instance, existing benchmark studies like [19, 55, 29] often suggest that 8-bit (weight-only or not) quantization is a relatively "safe" compression scheme. Although technically lossy, 8-bit models can often maintain strong task performance across a range of standard benchmarks. However, we must note these benchmarks typically focus on a narrow set of tasks (e.g., WikiText2 perplexity, MMLU, Commonsense Reasoning), and thus fail to offer a comprehensive view of real-world LLM usage, especially from the perspective of end-users.

That being said, the argument that "current benchmarks fail to capture the performance gap between 8-bit compressed and 16-bit uncompressed models" is itself constrained by the limitations of the current benchmarking landscape, making it difficult to produce abundant supporting evidence. Nonetheless, some reports have begun to highlight such gaps. For example, human evaluations on LLM Arena[1] show a notable performance drop between Llama-3.1-405B-Instruct [20] and its W8A8 counterpart (Llama-3.1-405B-Instruct-FP8), particularly under coding (1293 vs. 1277) and long-query (1282 vs. 1275) tasks. Similarly, quantizing DeepSeek-R1-Distill-Llama-70B [21] from 16 bits to 8 bits results in a 23.7% drop on GPQA (from 9.51% to 7.25%).[2] Furthermore, reasoning, a core capability of modern LLMs, appears especially sensitive to compression loss. Recent benchmark [39] reveals that quantizing DeepSeek-R1-Distill-Qwen-1.5B with 8-bit SmoothQuant [51] (for weight, attention, and KV cache) leads to an average 9.09% drop in reasoning tasks (48.82% to 44.29%) across datasets like AIME, MATH-500, GPQA-Diamond, and LiveCodeBench. We leave more evidence exploring the performance gap between 8-bit quantized and uncompressed model in Appendix H.

Although the broader question: "Which specific task, on which model, using which quantization technique, under what conditions, will lead to a noticeable drop compared to FP16/BF16?" is likely to remain open-ended simply due to the sheer amount of potential combinations. It is fair to say that lossy quantization introduces complexities that some end-users would prefer to avoid, since it creates uncontrolled variables that must be empirically stress-tested for each deployment scenario.

To eliminate this burden, DFloat11 offers a compelling alternative: delivering **100% identical performance to the original model, while consuming only ~70% of the memory footprint with throughput benefits**, which is a unique and practical offering for resource-constrained deployment settings.

## B    Extended Related Works

**Data Formats for Model Weights**    LLM weights are typically stored in compact floating-point formats such as FP16 or BFloat16 (officially stylized as *bfloat16*[3]). FP16 allocates 1 sign bit, 5 exponent bits, and 10 mantissa bits, whereas BFloat16 uses 1 sign bit, 8 exponent bits, and 7 mantissa bits. Compared to FP16, BFloat16 offers a wider dynamic range at the cost of precision, which improves numerical stability and mitigates overflow issues during training [17, 30].

Compressed data formats typically aim for lower bit-widths. For example, FP8—which comes in both E4M3 (4 exponent bits, 3 mantissa bits, plus 1 sign bit) and E5M2 configurations—has seen reasonable adoption in LLM training and development. Integer formats like INT8 have also been well explored, as in `LLM.int8()` [8] and its following works. Formats with a stronger emphasis on efficiency, such

---

[1] https://x.com/lmarena_ai/status/1835760196758728898
[2] https://huggingface.co/RedHatAI/DeepSeek-R1-Distill-Llama-70B-quantized.w8a8
[3] https://cloud.google.com/blog/products/ai-machine-learning/bfloat16-the-secret-to-high-performance-on-cloud-tpus

as FP4, INT4, NF4 [9], and AF4 [58], use only 4 bits. In this work, we primarily focus on formats with ≥8 bits, as benchmark literature [55, 19, 39] often suggests that 8-bit quantization results in negligible performance drop—though we show in Section A that this claim is likely skewed due to evaluation selectiveness and benchmark limitations.

**Lossless Model Compression**   While lossy model compression techniques such as pruning and quantization [14, 37, 15] have received widespread attention, lossless model compression remains a relatively underexplored area. Upon careful investigation, we identified roughly four prior works that have made meaningful efforts in this space. *Deep Compression* [22] is a foundational work, applying Huffman coding [28] to quantized CNN models and achieving an additional ∼22% compression gain for model checkpoints. *ZipNN* [25] extended this idea to language models, comparing its results to classic lossless compression tools such as zlib [10] and zstd[4] and demonstrated superior compression gains. However, this line of work—including their industry counterparts, such as ezm7[5]—is limited in that its efficiency gains only apply to storage (reducing the size of model checkpoints) but offer no benefits during inference. While such storage savings are meaningful in large-scale training settings—where frequent snapshotting and checkpoint rollbacks are needed [47]—they have limited impact for everyday LLM end-users. Model downloading is typically a one-time cost, so even if a model checkpoint is compressed by 50%, it only cuts the download time at most by half, presumably over the model's entire lifecycle of deployment. Furthermore, checkpoints are usually stored on disk, where terabytes of capacity are easily available, making up a much looser constraint compared to GPU HBM (High Bandwidth Memory); one of the main resource constraints during inference.

We argue that a lossless compression technique would be substantially more impactful if it could deliver efficiency gains during inference—particularly on GPU-based systems, which is the default setup for LLM serving. In this context, *NeuZip* [23] is the only prior work we identify that supports GPU inference. NeuZip applies entropy encoding with layer-wise decompression to maintain a reduced memory footprint throughout serving. However, it is built on NVIDIA's nvCOMP: *"a high-speed data compression and decompression library optimized for NVIDIA GPUs"*.[6] Unfortunately, nvCOMP is no longer open-source (only binary executables are available), which hinders future research. Moreover, we empirically find that nvCOMP's inference throughput and latency are significantly worse than our proposed DFloat11 kernel, resulting in a pipeline that trades memory efficiency for substantial inference overhead (see Figure 7).

Another work referencing NeuZip is *Huff-LLM* [59], which also aims to reduce memory costs while maintaining efficient inference. However, its contributions are specific to FPGA-like architectures and do not apply to GPUs. To the best of our knowledge, the DFloat data format we presented (and its respective kernel support in DFloat11) shall serve as the only GPU-inference-friendly data format with lossless compression benefits.

**Efficient LLM Inference**   LLMs are computationally intensive and resource-demanding, making the efficiency of LLM inference a key research focus [52]. FlashAttention [7] accelerates exact attention computation on GPUs through kernel fusion, while NoMAD Attention [64] speeds up attention on CPUs using in-register lookups. Model compression is another effective strategy to reduce resource requirements for serving LLMs and diffusion models. Quantization methods such as GPTQ [15], AWQ [37], SmoothQuant [51], LeanQuant [61], CQ [63], KVQuant [26], and KIVI [40] lower memory usage and enhance efficiency by compressing model weights, activations, or KV cache. Compression is also applied in fine-tuning: methods like LoRA [27], QLoRA [9], and SketchTune [62] compress model weight deltas, whereas GaLore [65] and SARA [60] compress optimizer states during training. One additional line of work relevant to efficient LLM inference would be *lossless efficient decoding*, where paradigms such as *speculative decoding* [49, 34, 50] and *n-gram candidate decoding* [16, 3] offer lossless generation quality with improved latency. DFloat11 mainly differs from these works in that it provides substantial savings in memory footprint while maintaining lossless generation quality, whereas most—if not all—lossless efficient decoding methods require memory consumption equal to or greater than that of the original model.

---

[4]https://github.com/facebook/zstd
[5]https://github.com/liuliu/s4nnc/pull/11          https://encode.su/threads/4067-Good-Compressors-for-16-bit-floats
[6]https://developer.nvidia.com/nvcomp

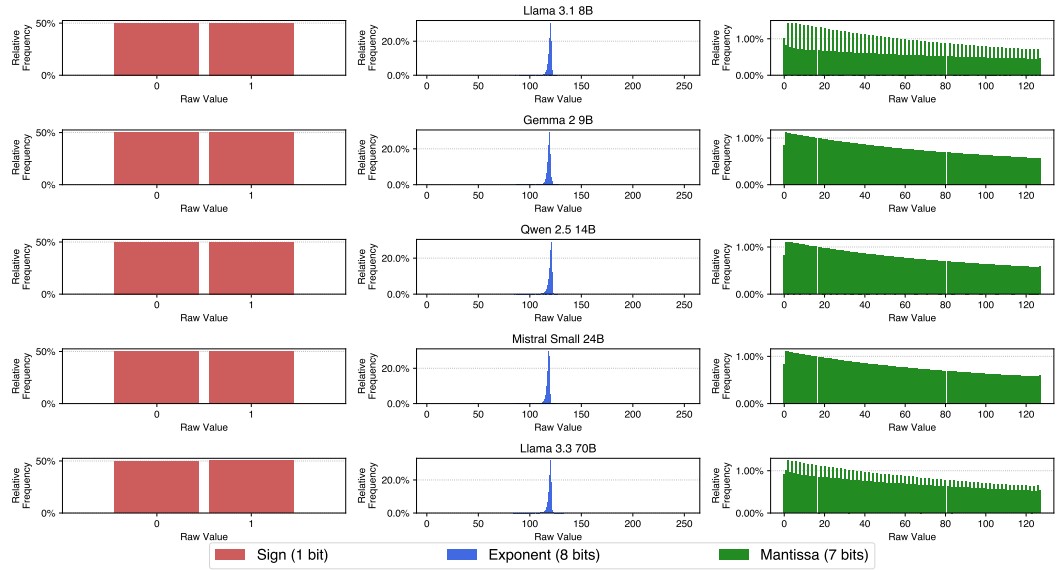

Figure 8: Relative frequency distribution of sign, exponent, and mantissa values in the BFloat16 weights of all linear projection layers across various LLMs.

## C  Frequency Distribution of BFloat16 Values

Figure 8 presents the frequency distribution for distinct values of sign, exponent, and mantissa bits in the BFloat16 weights of LLMs. Figure 9 shows the sorted frequency of exponent values of LLM weights.

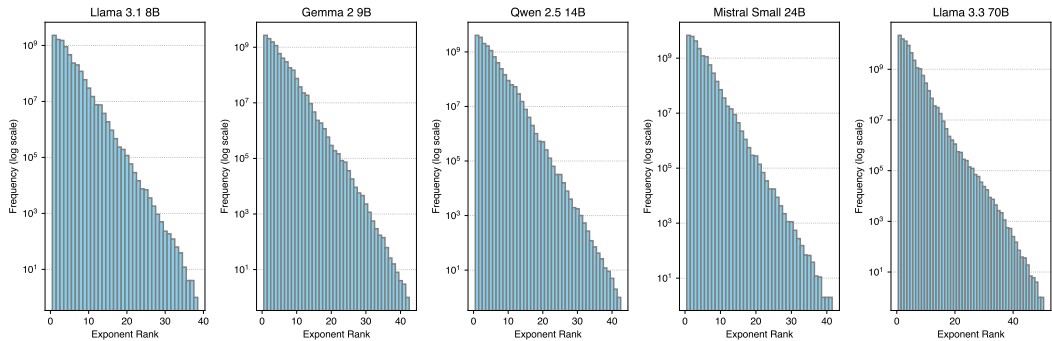

Figure 9: Distribution of BFloat16 exponent values across various models. The frequency of exponent values (shown in log scale) decays rapidly with exponent rank.

## D  Pseudo-code of the GPU kernel for DFloat11 Decompression

Algorithm 1 presents the pseudo-code of the two-phase GPU kernel for decompressing DFloat11 to BFloat16.

Table 4: System specifications of servers used for experiments.

|  | GPU | GPU Memory | CPU | CPU Memory |
|---|---|---|---|---|
| Server 1 | NVIDIA RTX A5000 | 24564MiB | AMD EPYC 7513 32-Core | 504GB |
| Server 2 | NVIDIA A100 | 40960MiB | AMD EPYC 7742 64-Core | 1.48TB |
| Server 3 | NVIDIA Quadro RTX 8000 | 49152MiB | AMD EPYC 7742 64-Core | 1.48TB |

**Algorithm 1** GPU kernel for decompressing DFloat11 to BFloat16

---

1: **procedure** DF11ToBF16
   **require:**
     – EncodedExponent, PackedSignMantissa: byte arrays
     – $\text{LUT}_1, \ldots, \text{LUT}_k$, CodeLengths: 8-bit unsigned integer arrays of size 256
     – Gaps: 5-bit unsigned integer array (one entry per thread in each block)
     – BlockOutputPos: 32-bit unsigned integer array (one entry per block)
     – Outputs: BFloat16 array, for storing results
     – $B, T, n, k$: the number of thread blocks, number of threads, number of bytes processed by each thread, number of compact LUTs, respectively
2:    Divide EncodedExponent into chunks:
     $\text{EncodedExponent}_1, \ldots, \text{EncodedExponent}_B$ of size $nT$ bytes each
3:    **for all** $b \leftarrow 1, \ldots, B$ **(in parallel across blocks) do**
4:      Load $\text{EncodedExponent}_b$ into SRAM
5:      Divide $\text{EncodedExponent}_b$ into chunks:
       $\text{EncodedExponent}_{b,1}, \ldots, \text{EncodedExponent}_{b,T}$ of size $n$ bytes each
6:      Load $\text{LUT}_1, \ldots, \text{LUT}_k$, CodeLengths into SRAM
7:      Initialize integer arrays $\text{NumElements}[1 \ldots T]$, $\text{ThreadOutputPos}[1 \ldots T]$ with all 0s
8:      Initialize BFloat16 write buffer WriteBuffer in SRAM
9:      **for all** $t \leftarrow 1, \ldots, T$ **(in parallel across threads) do**
       ▷ Phase 1: Each thread determines its initial output position
10:        $\text{BitOffset} \leftarrow \text{Gaps}[bT + t]$
11:        **while** $\text{BitOffset} < 8n$ **do**
12:          Read the next 4 bytes of $\text{EncodedExponent}_{b,t}$, starting from the BitOffset-th bit, into $\text{Byte}_{1\ldots 4}$
13:          $i \leftarrow 1$
14:          $\text{Exponent} \leftarrow \text{LUT}_1[\text{Byte}_i]$
15:          **while** $\text{Exponent} \geq 240$ **do**
           ▷ Exponent $\geq 240$ means that it is a pointer to the next LUT
16:            $i \leftarrow i + 1$
17:            $\text{Exponent} \leftarrow \text{LUT}_{(257-\text{Exponent})}[\text{Byte}_i]$
18:          **end while**
19:          $\text{BitOffset} \leftarrow \text{BitOffset} + \text{CodeLengths}[\text{Exponent}]$
20:          $\text{NumElements}[t] \leftarrow \text{NumElements}[t] + 1$
21:        **end while**
22:        **Thread Synchronization Barrier**
       ▷ Compute prefix-sum using Blelloch's Algorithm:
23:        $\text{ThreadOutputPos}[t] \leftarrow \text{BlockOutputPos}[b] + \sum_{i=1}^{t-1} \text{NumElements}[i]$
     ▷ Phase 2: Writing decoded BFloat16s to the appropriate positions
24:        $\text{BitOffset} \leftarrow \text{Gaps}[bT + t]$
25:        **while** $\text{BitOffset} < 8n$ **do**
26:          Read the next 4 bytes of $\text{EncodedExponent}_{b,t}$, starting from the BitOffset-th bit, into $\text{Byte}_{1\ldots 4}$
27:          $i \leftarrow 1$
28:          $\text{Exponent} \leftarrow \text{LUT}_1[\text{Byte}_i]$
29:          **while** $\text{Exponent} \geq 240$ **do**
           ▷ Exponent $\geq 240$ means that it is a pointer to the next LUT
30:            $i \leftarrow i + 1$
31:            $\text{Exponent} \leftarrow \text{LUT}_{(257-\text{Exponent})}[\text{Byte}_i]$
32:          **end while**
33:          $\text{Byte} \leftarrow \text{PackedSignMantissa}\big[\text{ThreadOutputPos}[t]\big]$
34:          $\text{Sign} \leftarrow \text{Byte} \ \texttt{bitwise\_and} \ \texttt{0b10000000}$
35:          $\text{Mantissa} \leftarrow \text{Byte} \ \texttt{bitwise\_and} \ \texttt{0b01111111}$
36:          $\text{WriteBuffer}[\text{ThreadOutputPos}[t] - \text{BlockOutputPos}[b]] \leftarrow$
           $(\text{Sign} \ \texttt{bitwise\_left\_shift} \ 8) \ \texttt{bitwise\_or}$
           $(\text{Exponent} \ \texttt{bitwise\_left\_shift} \ 7) \ \texttt{bitwise\_or} \ \text{Mantissa}$
37:          $\text{BitOffset} \leftarrow \text{BitOffset} + \text{CodeLengths}[\text{Exponent}]$
38:          $\text{ThreadOutputPos}[t] \leftarrow \text{ThreadOutputPos}[t] + 1$
39:        **end while**
40:      **end for**
     ▷ Perform coalesced writes to HBM:
41:      $\text{Outputs}[\text{BlockOutputPos}[b] \ldots (\text{BlockOutputPos}[b + 1] - 1)] \leftarrow$
       $\text{WriteBuffer}[0 \ldots (\text{BlockOutputPos}[b + 1] - \text{BlockOutputPos}[b] - 1)]$
42:    **end for**
43: **end procedure**

---

# E  Hardware for Experiments

Table 4 presents the hardware configuration of servers used for experiments.

# F  DFloat11 Compression Time

Table 5: Compression time per transformer block for different models.

| Model | Compression Time per Transformer Block (s) |
|---|---|
| Llama 3.1 8B Instruct | 191 |
| Llama 3.3 70B Instruct | 547 |
| Llama 3.1 405B Instruct | 2133 |

Table 5 reports the time required to compress a single transformer block for models of different sizes. Compression is a one-time preprocessing step for each model and is performed using a single CPU thread. Since transformer blocks are independent in terms of weight storage, their compression can be parallelized across multiple CPU threads, making the overall process highly scalable and efficient.

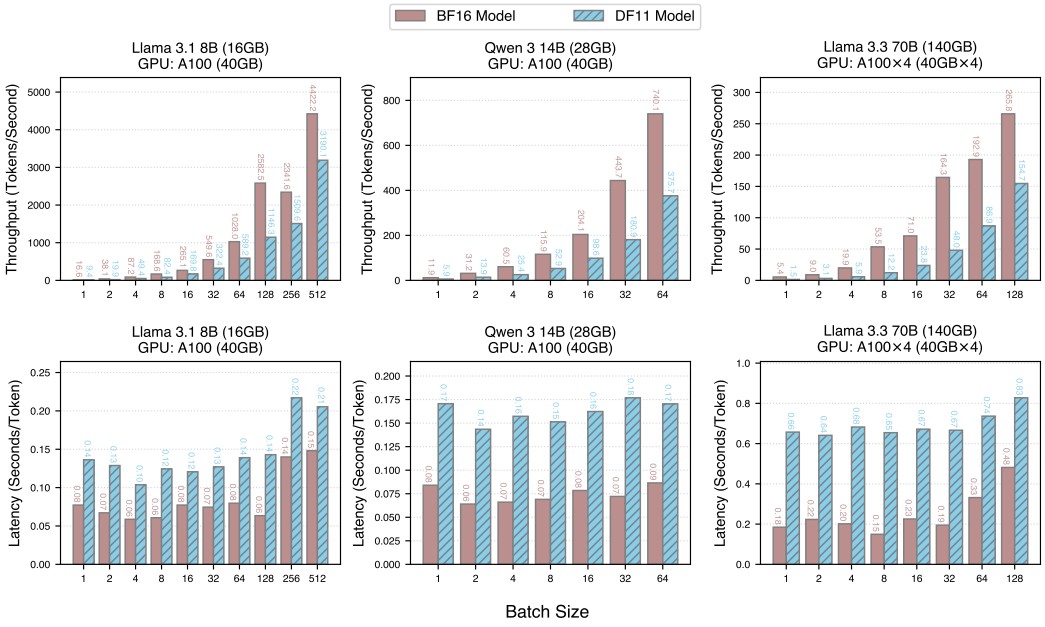

Figure 10: Comparison of average latency and throughput for token decoding between the original (BF16) models and their losslessly compressed (DF11) counterparts. The BF16 and DF11 models are run on the same GPU configurations, with Flash Attention [7] turned on for both methods.

# G  GPU Inference Efficiency Comparison: BF16 vs. DF11

We present the GPU inference efficiency of BF16 and DF11 models in Figure 10, for various models and batch sizes on A100 GPUs.

# H  Impact of Lossy Quantization

An accuracy comparison of the original and INT8-quantized Llama model is presented in table 6.

Table 6: INT8 quantization error on different tasks. "Math" denotes MATH Hard with 2 shots. "GPQA CoT" is with 2 shots. "Δ" denotes the error gap via INT8 quantization.

| Model | Data Type | Math | GPQA CoT |
|---|---|---|---|
| | BF16 | 23.92 | 15.18 |
| Llama-3.1-8B-Instruct | INT8 | 19.92 | 14.06 |
| | Δ | 4.0 | 1.12 |

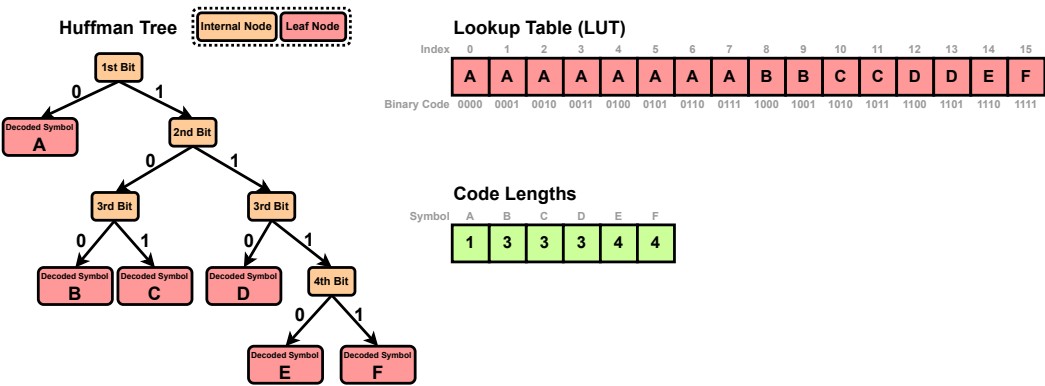

Figure 11: Decoding Huffman codes can be performed either by traversing the Huffman tree or by using two lookup tables: one that maps each $L$-bit binary code to its corresponding symbol, and another that stores the code length for each symbol.

# I  Efficient Decoding of Huffman Codes Using Compact Lookup Tables

## I.1  The Dual Lookup Table Approach

Huffman decoding can be performed by traversing the Huffman tree: starting from the root, each bit of the encoded bitstream determines the branch to follow, and the symbol is fully decoded upon reaching a leaf node. While this bit-by-bit traversal is conceptually simple, it is inefficient in practice. Each branching decision depends on the previous one, leading to frequent memory accesses and conditional jumps. This pattern is especially problematic on GPUs, where it causes branch divergence and limits instruction-level parallelism. A widely adopted alternative is *lookup-table-based decoding* [53], which flattens the Huffman tree into two compact lookup tables. This enables decoding of each symbol using just two array lookups and a bit shift, significantly improving throughput.

We employ two lookup tables, LUT and CodeLengths, to achieve efficient, branch-free Huffman decoding. Let $L$ denote the length of the longest codeword in the Huffman codebook. We construct the primary lookup table LUT as an array of size $2^L$, where each entry maps an $L$-bit binary sequence to the first symbol it encodes.

Figure 11 shows an example with $L = 4$ and a set of symbols A, B, C, D, E, F. For clarity, we use letters to represent symbols, though in practice these correspond to exponent values in BFloat16 weights. The lookup table LUT contains $2^4 = 16$ entries, indexed by all possible 4-bit binary sequences. Each entry in LUT stores the symbol whose Huffman code matches the prefix of that index. If a symbol's Huffman code is shorter than $L$ bits, it will fill multiple consecutive entries. For example, if symbol A is encoded as the single bit 0, then all binary sequences from 0000 to 0111 begin with 0, so entries 0 through 7 in LUT are assigned to A. In contrast, symbols with Huffman codes of length $L$ occupy exactly one entry each. For instance, E = 1110 and F = 1111 map to entries 14 and 15, respectively. This construction yields a dense prefix table that allows decoding a symbol with a single array lookup using an $L$-bit segment from the encoded bitstream.

To advance the encoded bitstream for decoding the next symbol, we also store the code lengths of all symbols. The second lookup table, CodeLengths, maps each symbol to its Huffman code length. In

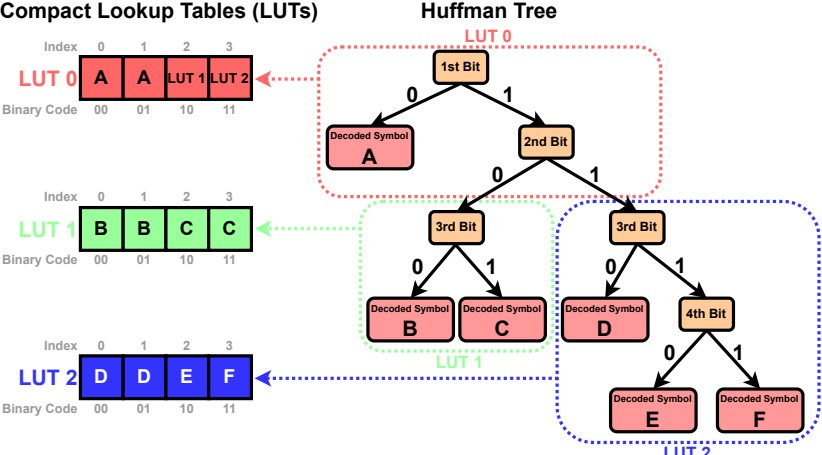

Figure 12: A Huffman tree can be decomposed into a hierarchy of subtrees, each represented by a compact lookup table (LUT). Each LUT may reference another lower-level LUT in the hierarchy. This hierarchical decoding approach is functionally equivalent to using a single monolithic LUT, but significantly more memory efficient.

the example, the lengths are: `A:1, B:3, C:3, D:3, E:4, F:4`. Together, these two tables allow fast, deterministic decoding by repeating the following steps:

1. Use the next $L$ bits from the encoded bitstream to index LUT and retrieve the decoded symbol.

2. Look up the code length of the decoded symbol from CodeLengths to determine how many bits to consume.

3. Advance the encoded bitstream and repeat.

This approach eliminates conditional branches and pointer chasing during decoding, making it highly suitable for parallel computation on GPUs.

### I.2   Decomposing LUT into Hierarchical, Compact Lookup Tables

The primary lookup table LUT contains $2^L$ entries, where $L$ is the maximum code length in the Huffman codebook. While this enables constant-time decoding, the table size grows exponentially with $L$. In practice, $L$ ranges from 24 to 32 for Huffman trees built with BFloat16 exponents. This results in table sizes of $2^{24}$ to $2^{32}$ entries, which far exceeds the capacity of GPU SRAM. To address this, we decompose LUT into multiple smaller lookup tables that fit within on-chip memory, while still enabling fast decoding.

**Hierarchical Table Structure**   Instead of storing a single flat table of size $2^L$, we decompose LUT into a hierarchy of compact lookup tables. Each table corresponds to a subtree of the Huffman tree and is responsible for decoding $b$ bits. Each table processes the next $b$ bits and either (i) directly returns a decoded symbol, or (ii) delegates to a table next in the hierarchy for decoding the next $b$ bits. This hierarchical organization mirrors the structure of the original Huffman tree and significantly reduces total memory usage.

Figure 12 illustrates an example where the Huffman tree is partitioned into three subtrees, each mapped to a separate lookup table responsible for 2 bits. The decoding process using these three LUTs proceeds as follows:

- $\text{LUT}_0$: Uses the first and second bits of the encoded bitstream to determine how to proceed, leading to 3 possible cases:
    - 00, 01 $\rightarrow$ decode the next symbol as A.
    - 10 $\rightarrow$ delegate to $\text{LUT}_1$ .

- 11 → delegate to $\mathsf{LUT}_2$.
- $\mathsf{LUT}_1$: Uses the third and fourth bits of the encoded bitstream to continue decoding:
    - 00, 01 → decode the next symbol as B
    - 10, 11 → decode the next symbol as C
- $\mathsf{LUT}_2$: Uses the third and fourth bits of the encoded bitstream to continue decoding:
    - 00, 01 → decode the next symbol as D
    - 10 → decode the next symbol as E
    - 11 → decode the next symbol as F

For decoding Huffman-coded BFloat16 exponents, we decompose the LUT into multiple compact lookup tables, each responsible for decoding 8 bits (i.e. $b = 8$). This allows us to read the next byte from the encoded bitstream and perform an array lookup from a 256-entry array in each step. In practice, the decomposition of LUT leads to 4 to 8 compact LUTs, each with 256 entries, which comfortably fits within fast SRAM.

## J  Text-to-image Results of BF16 and DF11 Diffusion Models

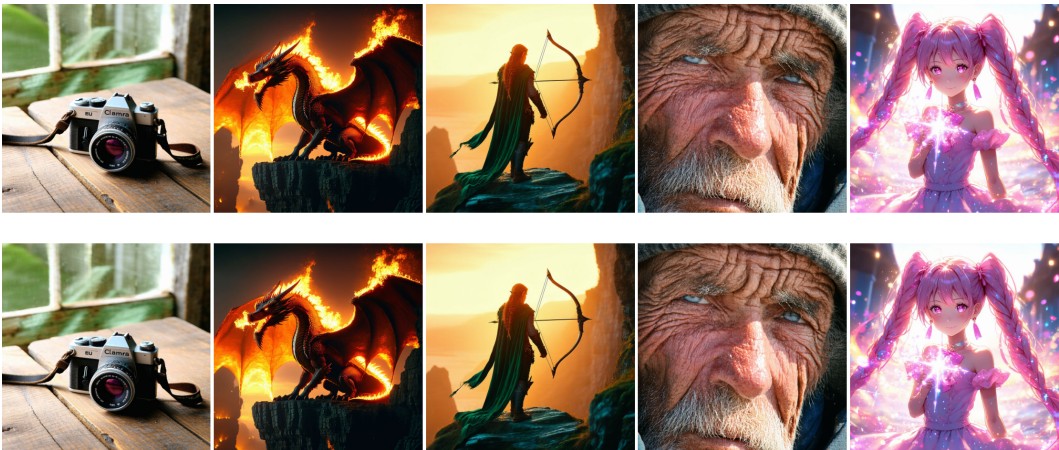

Figure 13: Images generated by Stable Diffusion 3.5 Large in the original BFloat16 precision **(top 5)** are pixel-wise identical to those produced by the DFloat11-compressed model **(bottom 5)**, using the same prompt and random seed.

Figure 13 presents the comparison of images generated using Stable Diffusion 3.5 Large in BFloat16 and DFloat11 weight format. The images are pixel-wise identical, when using the same prompt and random seed.

## K  Limitations

This work focuses exclusively on losslessly compressing BFloat16 weights. We do not consider other formats such as FP32, FP16, or FP8, which may require different compression strategies. While DF11 improves memory efficiency, it introduces a small but non-zero latency overhead due to decompression. This overhead is amortized at larger batch sizes but may impact latency-sensitive applications with small batches. Our evaluation is limited to GPUs. We do not assess performance on other hardware such as CPUs, TPUs, or custom accelerators, which may require platform-specific optimizations.

