# OpenReview forum: "70% Size, 100% Accuracy: Lossless LLM Compression for Efficient GPU Inference via Dynamic-Length Float (DFloat11)"
_NeurIPS.cc/2025/Conference — NeurIPS 2025 poster_

### Official Review · Reviewer_eUCL · 2025-07-02

**Clarity:** 3
**Significance:** 4
**Originality:** 4
**Rating:** 5
**Confidence:** 4

**Summary:**

This paper introduces Dynamic-Length Float (DFloat11), a lossless compression framework for BFloat16 model weights, achieving ~30% size reduction while preserving bit-for-bit accuracy. The method leverages entropy coding for exponents and employs optimized GPU kernels for efficient decompression during inference. Experiments on models like Llama 3 and Stable Diffusion demonstrate significant memory savings and throughput improvements.

**Questions:**

Please refer to the weaknesses.

**Ethical Concerns:**

["NO or VERY MINOR ethics concerns only"]

**Final Justification:**

Nice work. The overall results are very positive. The authors have also made a good rebuttal.

**Limitations:**

yes

**Quality:**

3

**Strengths And Weaknesses:**

## Strengths

1. Great insight. The authors identify the redundancy of BF16 and then propose to leverage entropy coding for exponents. The proposed DFloat11 maintains almost 100% model accuracy, which is impressive.

2. The authors implemented fast GPU kernels, which enables practical usages for on different hardwares.

3. Overall, the performance of this method is great. The paper is well-written and in a good shape.

## Weaknesses

1. My main concern is the evaluation on LLM is not very comprehensive. Specially, recent LLM releases tend to report performance across different tasks, including code, math, multilingual, instruction following, etc. It is not very clear how the proposed DFloat11 affect LLM's performance in these tasks. In my opinion, WikiText and C4 cannot fully represent the capability of today's LLMs.

2. Latency Overhead: Small but non-zero decompression costs may affect latency-sensitive applications, though amortized at larger batches.

---

> ### Author Rebuttal · Authors · 2025-07-31
>
> We sincerely thank the reviewer for the thoughtful feedback and for recognizing the insightfulness of our findings and the effectiveness of our approach. Below, we address the concerns raised in your review.
>
> **[W1] Evaluation on LLM is not very compreshensive.**
>
> We present additional evaluations of BF16 and DF11 models on various benchmarks in the table below. Since DF11 models produce bit-for-bit identical outputs to their BF16 counterparts, their performance on any benchmark is exactly the same. Figure 13 further illustrates the identical outputs generated by DF11 and BF16 models.
>
> | Model | MMLU  | ARC-C | Winogrande | BBH  | MuSR |
> |-|-|-|-|-|-|
> | Llama-3.1-8B-Instruct (BF16)       | 68.0 | 82.0  | 78.5       | 30.1 | 7.6  |
> | Llama-3.1-8B-Instruct (DF11)       | 68.0 | 82.0  | 78.5       | 30.1 | 7.6  |
>
> **[W2] Latency overhead**
>
> We agree that the non-zero decompression overhead may impact latency-sensitive applications at small batch sizes. However, as shown in Figure 6, this overhead becomes negligible in large batch size settings due to amortization. Additionally, we plan to explore further hardware-aware optimizations (such as fusing the decompression and GEMM kernels) in future work to reduce latency overhead.

---

> > ### Comment · Reviewer_eUCL · 2025-08-05
> > **Official comments from Reviewer**
> >
> > Thanks for the rebuttal. The new result have addressed my concerns. I have no more questions.

---

### Official Review · Reviewer_JfK6 · 2025-07-02

**Clarity:** 4
**Significance:** 3
**Originality:** 3
**Rating:** 5
**Confidence:** 4

**Summary:**

The paper proposes Dynamic-Length Float (DFloat11), a lossless compression framework that compresses weights of bfloat16-based LLMs and diffusion models to ~70% of their original size while preserving them bit-for-bit. The authors observe that the 8-bit exponent field in bfloat16 weights carries about less than 3 bits of information. They exploit this redundancy by Huffman-coding the exponent bits and leaving the sign and mantissa untouched, yielding an effective 11-bit representation. The weights are meant to be quickly decompressed with a GPU-based algorithm which stores decomposed LUTs in SRAM of every thread block, rendering the decompression operation fast and memory-bound.

**Questions:**

- Can DFloat11 be enabled on-demand only when the memory is exceeded (e.g., when the context gets too long)?
 - What would it take to fuse DF11 decompression into GEMM/attention kernels? How would this hypothetically lower the cost of decompression?

**Ethical Concerns:**

["NO or VERY MINOR ethics concerns only"]

**Final Justification:**

The problem under consideration is rather niche, but the contribution is focused, well engineered and well written. A strong factor in my recommended score is the author's promise to release the source code, so a broader audience could learn how to implement such a method in practice.

**Limitations:**

yes

**Paper Formatting Concerns:**

Minor: the headers do not follow the 2025 NeurIPS LaTeX template, which states: _All headings should be lower case (except for first word and proper nouns), flush left, and bold._

**Quality:**

4

**Strengths And Weaknesses:**

**Strengths**
 - The paper is clearly written and carefully checked even for minor errors.
 - It offers a clever solution to a rarely touched problem of lossless model compression.
 - The appendix is well-thought out and complements the paper with insightful details.

**Weaknesses**
 - The paper is missing a clear presentation of the overhead introduced by DFloat11, e.g., showing how large is the latency overhead in relation to BF16 calculation, when weight matrix dimensions and batch size change. This is somewhat visible _in vivo_ on Figure 6, which hints that for small token batch size the latency roughly doubles because DF11 is memory bound, and for large batch sizes it hides behind FLOPs. It would be useful to have a performance benchmark for, e.g., a single linear layer (of different sizes) and not an entire model.
- Quantization of bfloat16 models to > 8 bits is a niche problem, since proficient methods of quantizing to 8-bit formats or lower exist and the format is widely supported in hardware ; however, the authors well justify the lossless approach, which is a small but nonetheless valuable contribution.
 - Figure 4: The font on the plots is hard to read after printing. Perhaps a shared ylabel would save some space that could be spent there.
 - In general, the figures are hard to read, some fonts are too small.
 - Minor: it seems that the common capitalization (also used by Google which introduced this format) is “bfloat16”, while the paper uses “BFloat16”.
 - Source code absent (but promised to be released upon acceptance). In a paper like this, where the kernel is the central contribution, releasing the code is in my opinion a necessity.

---

> ### Author Rebuttal · Authors · 2025-07-31
>
> We thank the reviewer for giving insightful feedback and for highlighting the clarity and contribution of our work. We address the comments and questions below.
>
> **[W1] Performance benchmark for a single layer**
>
> Below, we provide additional performance benchmarks for a single transformer layer from Llama 3.1 8B Instruct, evaluated at various token batch sizes. Additional results for single layer decompression can be found in Figure 7, which presents the decompression throughput and latency for various matrices ranging from 4096x2048 to 4096x65536 in size.
>
> |Token Batch Size|BF16 (ms)|DF11 (ms)|Decompression Overhead (%)|
> |-|-|-|-|
> |1    |1.190    |2.414    |102.8       |
> |1024 |2.823    |4.361    |54.5        |
> |2048 |5.172    |6.736    |30.2        |
> |4096 |10.148   |11.687   |15.2        |
> |16384|45.436   |47.030   |3.5         |
> |32768|107.156  |108.478  |1.2         |
>
> **[W2] Quantization of bfloat16 models to more than 8 bits is a niche problem**
>
> Our work addresses a different goal from quantization methods. While quantization introduces a trade-off between accuracy and compression, our approach guarantees bit-identical outputs to the original bfloat16 model. This can be valuable in scenarios where exact numerical accuracy is critical or where the model's performance is highly sensitive to quantization.
>
> **[W3,W4,W5] Figure font size and the capitalization of bfloat16**
>
> We appreciate the reviewer's careful review of our paper. We will revise the figures in the camera-ready version to improve readability. We will also add a note to acknowledge the alternate capitalization of bfloat16.
>
> **[W6] Source code**
>
> We will release the source code upon paper acceptance.
>
> **[Q1] Can DFloat11 be enabled on-demand?**
>
> Yes, enabling DFloat11 on demand is feasible. Since our method compresses only the exponent bits while leaving the sign and mantissa unchanged, the compressed exponents can be offloaded to the CPU and transferred to the GPU to save memory when needed.
>
> **[Q2] Fusing DF11 decompression with GEMM**
>
> Fusing DF11 decompression into GEMM or attention kernels is a promising direction that could further reduce decompression overhead. This approach would involve grouping compressed exponent values into tiles or blocks, which each thread block can load and decompress in SRAM. The decompressed weights could then be used directly on chip for matrix multiplications, avoiding the bottleneck of an additional write and read of decompressed weights to and from HBM. We thank the reviewer for this insightful suggestion and plan to explore it in future work.

---

> > ### Comment · Reviewer_JfK6 · 2025-08-03
> >
> > Thank you for addressing my concerns and providing additional benchmarks. I will maintain my original score.

---

### Official Review · Reviewer_K1Gb · 2025-07-04

**Clarity:** 3
**Significance:** 2
**Originality:** 3
**Rating:** 5
**Confidence:** 5

**Summary:**

This paper introduces DF11, a novel data format to losslessly reduce the size of LLMs by applying entropy coding. DFloat11 assigns dynamic length encoding to weights. The authors also write custom GPU kernels for fast online decompression. This method shows improvement over a large range of models, improving efficiency, generation length and the capability of GPU hosting LLMs.

**Questions:**

1. Is it possible to apply the dynamic length weight to FP8/FP4 models? What is the room for improvement?
2. How long does it take to compress LLMs in DF11? For example, 1B/8B/32B/405B.
3. How is the latency when batch size is 1/2/4/8/16/32?

**Ethical Concerns:**

["NO or VERY MINOR ethics concerns only"]

**Final Justification:**

While the proposed algorithm is interesting, the demonstrated performance gain is limited (only 30% model compression). Although it is a lossless method, it does not seem to bring significant performance gain compared with FP8/FP4, especially in the context that lower bits computation by hardware is growing every year.

**Limitations:**

yes.

**Paper Formatting Concerns:**

No.

**Quality:**

4

**Strengths And Weaknesses:**

Strenghs
1. The idea of losslessly compressing LLMs is interesting and novel.
2. The experiments are convincing and detailed.
3. The writings are easy to follow.
4. The algorithms to support the lossless compress make sense and are hardware-efficient.

Weeknesses
1. The paper does not show the performance gap between DF11 and AWQ/GPTQ/FP8, even if I acknoledge that these methods are lossy.
2. The paper does not provide quantitative information on the time consumption of compression.
3. DFloat11 is less helpful when FP8 becomes the default data format. (e.g., DeepSeek-V3/R1)
4. Token Batch Size 16384 is not that convincing, since I assume the use case of DF11 is when VRAM is limited. That means this token batch size should refer to prefilling stage. However, except for super-long context, (where KV cache memory will become the major bottleneck), the major overhead is in decoding. Assuming 40GB A100 and Llama3.1-8B, the common serving batch size is hardly beyond 16-32. In this regime, the parameter loading will become the main cost. DF11, to my understanding, will at least cost 70% additional overhead since it will load the 11-bit and 16-bit weights.

---

> ### Author Rebuttal · Authors · 2025-07-31
>
> We thank the reviewer for providing valuable feedback and for highlighting the novelty of our work and the effectiveness of our method. We address your concerns and questions below.
>
> **[W1] Performance comparison between DF11 and AWQ/GPTQ/FP8**
>
> We present additional accuracy comparisons between BF16, DF11, FP8, and GPTQ (INT4). As shown in the table below, FP8 and GPTQ introduce non-zero accuracy degradation on various benchmarks, while DF11 maintains identical accuracy to BF16. Moreover, even in cases where the accuracy remains roughly unchanged, the quantized model's behavior can diverge significantly from that of the original model, as discussed in Line 44 of the paper.
>
> | Model | MMLU  | ARC-C | Winogrande | BBH  | MuSR |
> |-|-|-|-|-|-|
> | Llama-3.1-8B-Instruct (BF16)       | 68.0 | 82.0  | 78.5       | 30.1 | 7.6  |
> | Llama-3.1-8B-Instruct (DF11)       | 68.0 | 82.0  | 78.5       | 30.1 | 7.6  |
> | Llama-3.1-8B-Instruct (FP8)             | 68.0 | 81.2  | 77.7       | 29.7 | 7.5  |
> | Llama-3.1-8B-Instruct (INT4,GPTQ)      | 66.9  | 80.2  | 78.0       | 28.9 | 6.3  |
>
> We also provide throughput results (prefill 2K tokens, decode 1K tokens) for DF11, BF16, and GPTQ (INT4) with different batch sizes below. These results show that while DF11 incurs additional decompression overhead, it remains practical across batch sizes. Importantly, DF11 guarantees accuracy and behavior that are exactly identical to the original model, whereas quantized models (such as FP8, GPTQ, AWQ, etc.) may experience accuracy loss or behavioral changes.
>
> | Model  | BS 1 | BS 4 | BS 16 | BS 64 |
> |-|-|-|-|-|
> | Llama-3.1-8B-Instruct (BF16) | 27.51 | 108.80  | 494.20  | 1883.91 |
> | Llama-3.1-8B-Instruct (DF11) | 13.17 |  51.91  | 203.64  |  774.52 |
> | Llama-3.1-8B-Instruct (INT4,GPTQ) | 32.01 | 126.72  | 502.89  | 1929.24 |
>
> **[W2, Q2] Time consumption of compression**
>
> We report detailed compression times in Appendix F. Compressing an 8B model takes approximately 5 minutes, while a 405B model requires about 4 hours. Compression is performed only once per model and is fast and efficient. Since each transformer block can be compressed independently, we parallelize the process across multiple CPU cores to further accelerate compression.
>
> **[W3,Q1] Applying DFloat to FP8/FP4 models**
>
> Dynamic-length float can be applied to FP8 and FP4 weights to enable lossless compression for these formats. Similar to BF16, we apply Huffman coding to compress the exponent bits while leaving the sign and mantissa uncompressed. Since FP4 and FP8 have fewer exponent bits than BF16, the resulting lookup tables are smaller, which may allow for faster decompression. The compression gains vary across these formats, as the effectiveness depends on the entropy of the weights. For example, we measured the entropy of FP8 weights in Qwen3-8B to be 6.78 bits, corresponding to a 15.3% compression ratio. In contrast, FP4 weights in Llama-3.3-70B have an entropy of 3.97 bits (a 0.8% compression ratio), likely too small to be impactful. We thank the reviewer for this suggestion and plan to explore it further in future work.
>
> **[W4] Relevance of Large Token Batch Sizes and Decompression Overhead**
>
> While we agree that typical serving batch sizes for decoder-only LLMs are usually limited to 16-32, larger token batch sizes are relevant in other important scenarios. For example, when using LLMs as embedding models (where no KV cache is involved) or in diffusion models for image and video generation, the models process a large number of tokens at once. In these settings, the decompression overhead is effectively amortized and becomes negligible. As shown in Table 3, DF11 inference latency for diffusion models increases by only 4-5% compared to BF16.
>
> For smaller batch sizes (e.g., 16–32), the additional overhead is not a fixed 70%. Instead, it is the constant overhead required to decompress the model from DF11 to BF16. Figure 10 provides a detailed comparison of inference efficiency between BF16 and DF11 across various batch sizes and models.
>
> **[Q3] Latency for different batch sizes**
>
> Please refer to Figures 4 and 10 for latency measurements across batch sizes of 1, 2, 4, 8, 16, and 32. As the batch size doubles, throughput approximately doubles while latency increases slightly.

---

> > ### Comment · Reviewer_K1Gb · 2025-08-05
> >
> > While the proposed algorithm is interesting, the demonstrated performance gain is limited (only 30% model compression). Although it is a lossless method, it does not seem to bring significant performance gain compared with FP8/FP4, especially in the context that lower bits computation by hardware is growing every year.  Besides, I am supportive for this acceptance.

---

### Official Review · Reviewer_uiTc · 2025-07-09

**Clarity:** 4
**Significance:** 4
**Originality:** 3
**Rating:** 5
**Confidence:** 4

**Summary:**

The paper introduces Dynamic-Length Float (DFloat11), a lossless compression method for storing BF16 model weights by applying Huffman coding only to the exponents. This reduces the original 8-bit exponent representation to an average of 2.6 bits. Additionally, the paper proposes an efficient GPU-friendly implementation of Huffman en/decoding, addressing parallelization challenges to maintain acceptable inference speed.

**Questions:**

- Since gap arrays, index storage, and Huffman encoding are not novel techniques, what do you consider the key contribution or novelty of your paper?

- Could the proposed exponent-only compression strategy be extended to other formats such as FP16, FP8, or even 8-bit quantized models? If so, what kind of savings or trade-offs would you expect in those settings?

**Ethical Concerns:**

["NO or VERY MINOR ethics concerns only"]

**Quality:**

3

**Strengths And Weaknesses:**

**Strengths**
- The core idea of compressing exponents—given their lower entropy—is well-motivated and supported by entropy analysis across multiple models, making the approach convincing.

- The authors thoughtfully address implementation challenges, particularly for GPUs, and their proposed two-phase kernel to handle Huffman coding appears sound.

- The paper provides solid experimental results. End-to-end evaluations on large LLMs demonstrate that DFloat11 delivers real-world improvements.

**Weaknesses**
- Some of the proposed implementation optimizations for Huffman encoding—such as using a gap array and index storage—have already been explored in prior work, which may make these contributions appear incremental.

- The method is BF16-specific, while many production systems are shifting toward INT8, FP8, or mixed-precision formats. Although the paper mentions this limitation in the introduction, it lacks a more detailed discussion comparing BF16 accuracy and throughput with INT8, which would help clarify DFloat11’s relevance in current deployment contexts.

---

> ### Author Rebuttal · Authors · 2025-07-31
>
> We thank the reviewer for providing thoughtful feedback, and for recognizing the motivation of our work and the effectiveness of our proposal. We address your concerns and questions below.
>
> **[W1, Q1] Novelty and Contributions**
>
> While Huffman coding optimizations for GPUs have been studied by previous works, lossless model compression that supports GPU inference is relatively unexplored. Most previous work has focused on lossy compression (such as quantization) or lossless methods intended only for reducing storage on disk, rather than for use during inference. Supporting GPU inference of losslessly compressed models requires non-trivial algorithmic designs and engineering efforts. Our key contributions include:
> 1.  Practical solution for deployment: We bridge the gap between the low-entropy observation and real-world deployment by designing a practical method for online inference using losslessly compressed weights. This includes proposing transformer-block-level decompression to achieve better throughput on GPUs.
> 2.  Proven benefits: Our results demonstrate that significant compression gains can be achieved with minimal loss in GPU inference speed, while preserving losslessness.
>
> We believe our work opens up new directions for lossless model compression in online inference and can inspire further progress in efficient model deployment.
>
> **[W2] Comparison with INT8 Accuracy and Throughput**
>
> We have added additional accuracy comparisons between BF16, DF11, and INT8 models. The INT8 models use GPTQ-W8A8 quantization with the llm-compressor library. Our results show that BF16 and DF11 achieve identical accuracy across all tasks. In contrast, the INT8 models show a non-zero drop in accuracy compared to BF16/DF11, which can be as high as 6.37% on GSM8K with quantized Gemma-3-27b-it. We will include further throughput comparisons between BF16, DF11, and INT8 models in the final version of the paper. According to the SmoothQuant paper, INT8 quantization can achieve up to 1.35x to 1.42x speedup over 16-bit inference.
> |Model|MMLU|ARC Challenge|GSM8K|Winogrande|
> |-|-|-|-|-|
> |Gemma-3-27b-it (BF16/DF11)|77.5|72.5|92.1|79.4|
> |Gemma-3-27b-it (INT8)|76.4|70.8|85.8|79.7|
> |Llama-3.1-8B-Instruct (BF16/DF11)|68.0|82.0|82.8|78.5|
> |Llama-3.1-8B-Instruct (INT8)|67.8|81.7|84.8|78.5|
> |Llama-3.3-70B-Instruct (BF16/DF11)|86.6|49.2|94.2|84.8|
> |Llama-3.3-70B-Instruct (INT8)|85.9|48.0|94.0|83.7|
>
> **[Q2] Extension to Other Formats (FP16, FP8, and 8-bit Quantized Models)**
>
> Our method is also applicable to FP16, FP8, and other 8-bit quantized formats. The table below shows the entropy of Qwen3-8B weights in FP16, FP8, and INT8 formats, along with their corresponding compression factors. Dynamic-length float encoding can be extended to support these formats to enable their lossless compression.
>
> |Format|Bit Width|Entropy|Compression Factor (entropy/bitwidth)|
> |-|-|-|-|
> |FP16|16|13.69|85.6%|
> |FP8|8|6.78|84.8%|
> |INT8|8|7.50|93.8%|

---

### Decision · Program_Chairs · 2025-09-17

**Decision:**

Accept (poster)

**Comment:**

This paper introduces DF11, a novel data format to losslessly reduce the size of LLMs by applying entropy coding. DFloat11 assigns dynamic length encoding to weights. All reviewers agree that this work makes solid contributions, and the AC recommends acceptance.